# Enhancing electrochemical carbon dioxide capture with supercapacitors

Zhen Xu [1], Grace Mapstone[1], Zeke Coady[1], Mengnan Wang[2], Tristan L. Spreng[1], Xinyu Liu [1], Davide Molino[1,3] & Alexander C. Forse [1] ✉

Supercapacitors are emerging as energy-efficient and robust devices for electrochemical $CO_2$ capture. However, the impacts of electrode structure and charging protocols on $CO_2$ capture performance remain unclear. Therefore, this study develops structure-property-performance correlations for supercapacitor electrodes at different charging conditions. We find that electrodes with large surface areas and low oxygen functionalization generally perform best, while a combination of micro- and mesopores is important to achieve fast $CO_2$ capture rates. With these structural features and tunable charging protocols, YP80F activated carbon electrodes show the best $CO_2$ capture performance with a capture rate of 350 $mmol_{CO_2}$ $kg^{-1}$ $h^{-1}$ and a low electrical energy consumption of 18 kJ $mol_{CO_2}^{-1}$ at 300 mA $g^{-1}$ under $CO_2$, together with a long lifetime over 12000 cycles at 150 mA $g^{-1}$ under $CO_2$ and excellent $CO_2$ selectivity over $N_2$ and $O_2$. Operated in a "positive charging mode", the system achieves excellent electrochemical reversibility with Coulombic efficiencies over 99.8% in the presence of approximately 15% $O_2$, alongside stable cycling performance over 1000 cycles. This study paves the way for improved supercapacitor electrodes and charging protocols for electrochemical $CO_2$ capture.

Anthropogenic carbon dioxide ($CO_2$) emissions are the primary contributor to global warming and climate change, posing significant challenges to our world[1]. Traditional $CO_2$ capture methods using amine-based solvents or solid sorbents require energy-intensive temperature swing regeneration processes, which suffer from low energy efficiencies and short material lifetimes[2]. Electrochemical $CO_2$ capture is an emerging decarbonization approach that uses the charging and discharging of an electrochemical cell to drive $CO_2$ capture and release[3]. This approach employs electricity as the sole driving force and has the potential to become an energy-efficient and low-cost method to capture $CO_2$ at room temperature[3]. A range of electrochemical $CO_2$ capture technologies are under development, including those based on electrochemically-driven pH swings[4–6], redox-active $CO_2$-binding molecules[7–10], and electrochemically mediated amine

regeneration[11–13]. Key challenges for these electrochemical approaches include the use of critical materials[14], cell degradation[6], $O_2$ sensitivity[10], and low $CO_2$ capture capacities and rates[15].

Among the various electrochemical $CO_2$ capture technologies, aqueous supercapacitors appear promising due to their use of low-cost, abundant and sustainable materials, long cycle lifetimes, high energy efficiencies and fast charging kinetics[16]. These devices reversibly capture $CO_2$ when charged through an effect known as supercapacitive swing adsorption (SSA)[16]. The device configuration features a symmetric supercapacitor cell with two identical porous activated carbon electrodes and an aqueous electrolyte, and the cell is contacted with a $CO_2$-containing gas at one electrode (Fig. 1)[17]. While the exact molecular mechanism of $CO_2$ capture by these systems remains under investigation, the mechanism of capture likely involves charging-

[1]Yusuf Hamied Department of Chemistry, University of Cambridge, Cambridge, United Kingdom. [2]Department of Chemical Engineering, Imperial College London, London, United Kingdom. [3]Politecnico di Torino, Dipartimento di Scienza Applicata e Tecnologia (DISAT), Corso Duca degli Abruzzi, 24, Torino, Italy. ✉e-mail: acf50@cam.ac.uk

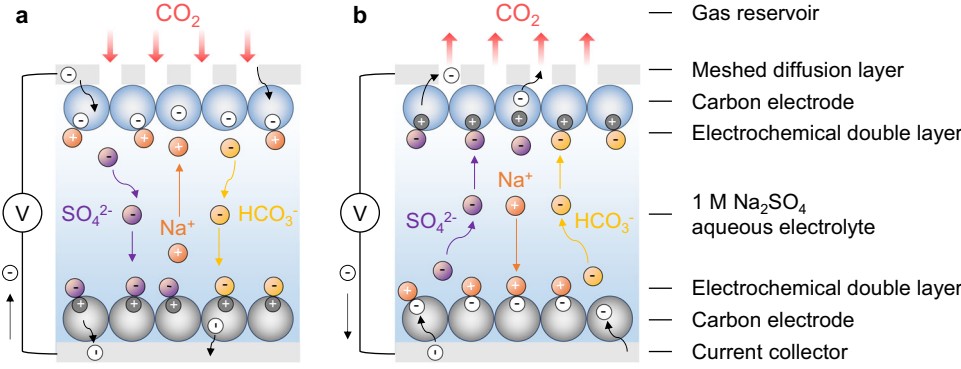

**Fig. 1 | Schematic diagram illustration of supercapacitors for electrochemical $CO_2$ capture.** The mechanistic hypothesis for **a** capture and **b** release of $CO_2$ and corresponding ion movements upon charging and discharging.

driven perturbations to the below equilibria[17]:

$$CO_2(g) \rightleftharpoons CO_2(aq) \qquad (1)$$

$$CO_2(aq) + H_2O(l) \rightleftharpoons H_2CO_3^{*}(aq) \qquad (2)$$

$$H_2CO_3^{*}(aq) + H_2O(l) \rightleftharpoons HCO_3^{-}(aq) + H_3O^{+}(aq) \qquad (3)$$

$$HCO_3^{-}(aq) + H_2O(l) \rightleftharpoons CO_3^{2-}(aq) + H_3O^{+}(aq) \qquad (4)$$

In our recent study, we found that when the gas-exposed working electrode was charged negatively, $CO_2$ capture was observed, while when the gas-exposed working electrode was charged positively, $CO_2$ release was observed (Fig. 1)[17]. We hypothesized that the negative charging process causes bicarbonate ions ($HCO_3^{-}$) to migrate away from the gas-exposed working electrode and leads to a local depletion of $CO_2$ which then drives $CO_2$ capture (Fig. 1a). In contrast, we proposed that the positive charging process accumulates $HCO_3^{-}$ ions on the gas-exposed working electrode and thereby drives $CO_2$ release (Fig. 1b).

Regardless of the operating mechanism, supercapacitors have significant potential advantages for electrochemical $CO_2$ capture (*e.g.*, long cycle lifetimes, fast charging kinetics, high round-trip energy efficiencies, etc.) compared with battery-type electrochemical $CO_2$ capture devices[3]. However, one challenge for this technology is the relatively low $CO_2$ adsorption capacities that are obtained (~100 mmol $CO_2$ per kilogram of the electrode), and it also remains unknown whether these devices can tolerate the presence of $O_2$, which often causes side reactions and degradation in electrochemical $CO_2$ capture devices. Recent works have begun to optimize electrode materials[18], electrolyte compositions[19,20] and charging protocols[17,21,22]. Most notably, a recent study showed that activated carbon electrodes with larger electrochemical capacitances could achieve larger $CO_2$ capture capacities with $CO_2$ capture rates approaching 300 $mmol_{CO2}$ $kg^{-1}$ $h^{-1}$ [18]. Despite this progress, it remains unclear how the specific electrode structure (*e.g.*, surface area, pore size, surface functional groups, etc.) correlates with electrochemical $CO_2$ capture performance at different charging conditions, making it challenging to design improved electrodes and charging protocols, which fully realize the potential of this system.

In this study, we investigate the relationship between electrode structure and electrochemical $CO_2$ capture performance for a range of charging protocols. We find that carbons with larger surface areas and electrochemical capacitances generally have larger $CO_2$ capture capacities, and a combination of micro- and mesopores is crucial for

attaining good kinetic performance of $CO_2$ capture, particularly at fast charging rates. In addition, the oxygen functionalization is unfavorable for the thermodynamic performance of $CO_2$ capture. YP80F, a biowaste-derived activated carbon with a high surface area, a combination of micro- and mesopores, and low oxygen functional group content, demonstrates the best $CO_2$ capture performance. By fine-tuning the testing parameters, our device achieves a high $CO_2$ adsorption capacity of 170 mmol $CO_2$ per kg of the electrode (30 mA $g^{-1}$, from −0.8 to +0.8 V, pure $CO_2$), a high adsorption rate of 350 mmol of $CO_2$ per kg of the electrode per hour (300 mA $g^{-1}$, 0.8 V, pure $CO_2$), very low electrical energy consumption less than 20 kJ per mol of adsorbed $CO_2$ (300 mA $g^{-1}$, 0.8 V, pure $CO_2$), no measurable degradation over 12000 cycles (150 mA $g^{-1}$, 0.8 V, pure $CO_2$) and excellent $CO_2$ selectivity over $N_2$ and $O_2$. Importantly, we find that oxygen reduction reactions can be suppressed by operating supercapacitor devices in a "positive charging mode", and we observe stable cycling performance for at least 1000 cycles with Coulombic efficiencies over 99.8% under mixed gas conditions (~20% $CO_2$, 15% $O_2$ and 65% $N_2$). Combined with the observed low electrical energy consumption values, this work shows the potential to enhance electrochemical $CO_2$ capture with supercapacitors.

## Results
A gas cell setup was first designed to enable simultaneous electrochemistry and $CO_2$ uptake measurements for electrode performance evaluation (Fig. 2a). To enable reproducible cell assembly and reduce internal resistances, supercapacitors were prepared in commercial coin cells with a meshed top case to allow for contact between the top working electrode and the gas reservoir, thus avoiding issues with high cell resistances in our previous Swagelok cell setup[17]. The set-up used in this work employs a static gas atmosphere with a fixed volume, where gas sorption during cell charging is monitored using a pressure sensor (see "Methods"). This set-up allows us to observe reversible gas uptake, as well as irreversible gas consumption to gain insights into possible electrode or device degradation (Supplementary Fig. 1).

### Effects of electrode charging protocols on electrochemical $CO_2$ capture
First, to investigate the impacts of electrode charging protocols on electrochemical $CO_2$ capture, we employed a standard activated carbon, YP50F (Fig. 2b), as a benchmark electrode material with a 1 M $Na_2SO_4$ (aq) electrolyte. In these experiments, the electrochemical cell was charged to a cell voltage of −0.8 V ("negative charging mode", the gas-exposed working electrode is negatively charged) or +0.8 V ("positive charging mode", the gas-exposed working electrode is positively charged) with constant current charging (and discharging) and 5-min voltage holds between the charge and discharge steps.

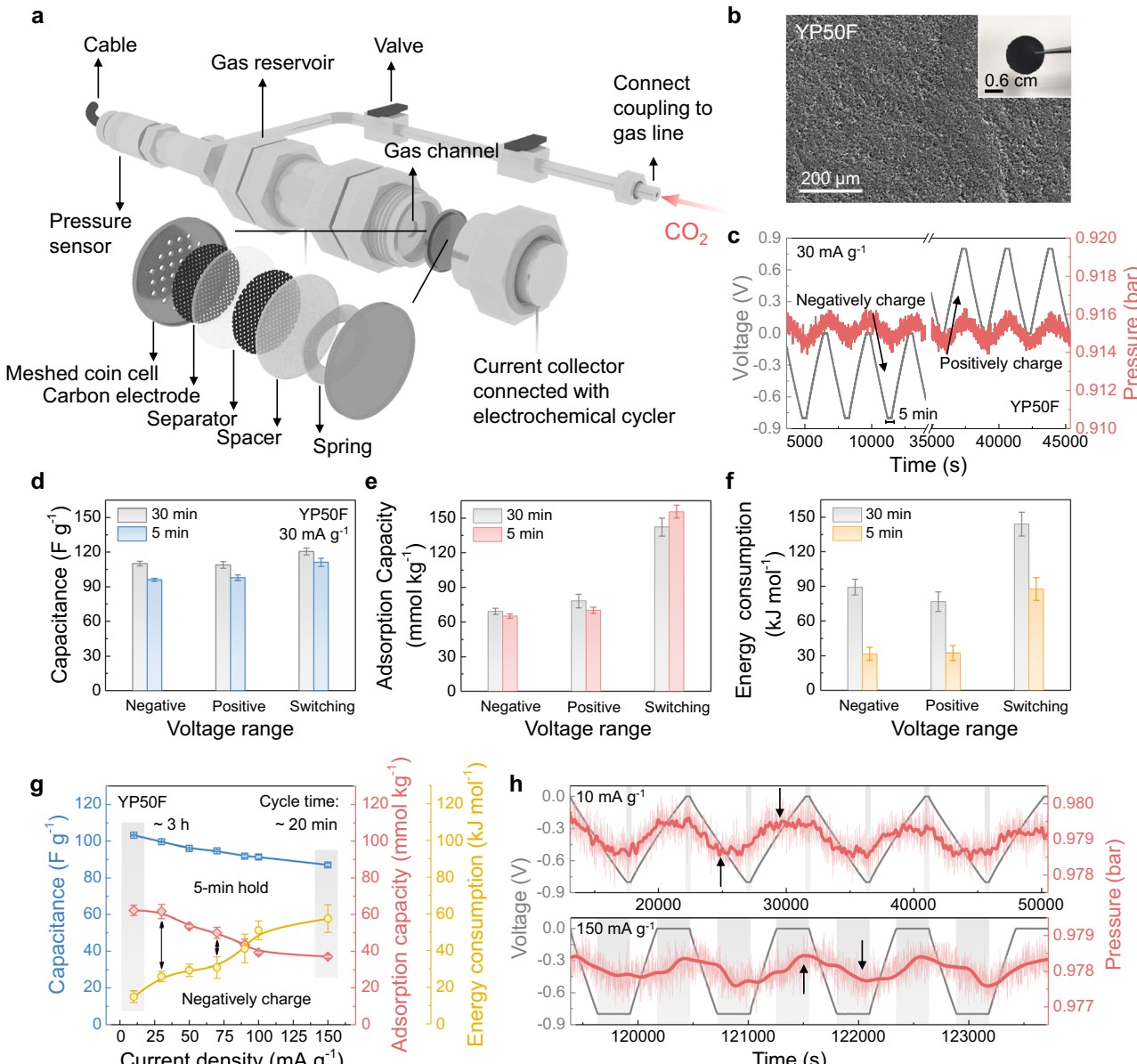

**Fig. 2 | Effects of electrode charging protocols on electrochemical CO₂ capture.** **a** Schematic of the custom-made gas cell setup that houses a meshed coin cell for electrochemical $CO_2$ capture measurements at 303 K. **b** Scanning electron microscopy (SEM) image of the YP50F electrode film, inset: a photo of a YP50F electrode. **c** Overall GCD curves (grey) and corresponding pressure curves (red) of the device using YP50F electrodes and 1 M Na₂SO₄ (aq) electrolyte for $CO_2$ sorption in negative and positive charging modes (under $CO_2$, at the current density of 30 mA g⁻¹, with a 5-min voltage hold after the charge or discharge process). Comparison of **d** the discharge capacitance, **e** $CO_2$ adsorption capacity, and **f** electrical energy consumption of the YP50F electrode under $CO_2$ at the current density of 30 mA g⁻¹ in different charging modes (i.e., negative, positive and switching charging modes), with different voltage hold time of 5 and 30 min. **g** Comparison of the discharge

capacitance, $CO_2$ adsorption capacity, and electrical energy consumption of the YP50F electrode under $CO_2$ at different current densities in the negative charging mode, with 5-min voltage holds (gray regions highlight the performance metrics at 10 and 150 mA g⁻¹, and black arrows represent the relationship between the $CO_2$ adsorption capacity and energy consumption). **h** Zoomed GCD curves (grey), original pressure curves (light red), and smoothed pressure curves (averaged every 100 sec, dark red) of the YP50F electrode under $CO_2$ at different current densities of 10 and 150 mA g⁻¹ in the negative charging mode, with 5-min voltage holds (gray regions represent the voltage hold steps, and black arrows represent the maximum and minimum peaks of pressure curves). Error bars represent the t-test of performance from cycle to cycle at the same charging protocol.

According to the three-electrode measurement of the symmetric supercapacitor with YP carbons, the cell voltage of 0.8 V is equally divided between the two electrodes (with respect to a reference electrode (Supplementary Fig. 2)). This voltage window was selected to avoid water splitting[22]. Similar to previous studies[16–18,22], in the galvanostatic charge and discharge (GCD) curves (Fig. 2c), $CO_2$ capture is observed upon negative charging to −0.8 V, evidenced by a decrease in the $CO_2$ gas pressure. Upon cell discharging back to 0 V, $CO_2$ release is then observed, with the behavior repeatable over several cycles. In

contrast, for the positive charging mode, $CO_2$ desorption is observed during charging, followed by $CO_2$ adsorption upon discharging (Fig. 2c). In summary, $CO_2$ adsorption is observed when the gas-exposed working electrode obtains electrons, while $CO_2$ desorption is observed when the working electrode loses electrons. These findings are very similar to our previous study of the same electrode with a 1 M NaCl (aq) electrolyte[17]. Compared to other electrochemical approaches using pH swings driven by proton-coupled electron transfer[4–6], or redox-active molecules that bind $CO_2$ upon electrochemical

reduction[7–10], supercapacitors offer the flexibility to capture $CO_2$ in both positive and negative charging modes.

To investigate the impacts of the voltage hold time between charge and discharge processes, experiments were performed with 5 min and 30 min voltage holds (Supplementary Fig. 3 and 4). In each case, positive and negative charging modes were studied, as well as a switching mode where the cell voltage was alternated between +0.8 and −0.8 V (Supplementary Figs. 3 and 4), an approach that was previously found to give higher capacities[17]. The electrode discharge capacitance slightly increases with increasing voltage hold time for all charging modes (Fig. 2d), and the $CO_2$ adsorption capacity only shows very minor changes (Fig. 2e). However, the electrical energy consumption significantly increases with the 30 min voltage holds for all charging modes (Fig. 2f). At the same time, the cycle time is inherently longer, leading to lower $CO_2$ adsorption rates. It is worth noting that fully removing the voltage hold leads to lower $CO_2$ adsorption capacities, especially at high current densities, which will be further discussed in the below section regarding practical applications. Overall, for a structure-property-performance study, the 5-min voltage holds are preferable. Similar to our previous study[17], we find that in the switching charging mode, the $CO_2$ adsorption capacity is almost doubled, indicating that the $CO_2$ adsorption capacity is related to the change in the number of charges carried by the electrode. The increased $CO_2$ adsorption in the switching mode comes at the cost of increased electrical energy consumption.

Furthermore, a series of experiments were performed to investigate the impacts of the current density during charging and discharging for the negative charging mode (Fig. 2g). The electrode discharge capacitance and $CO_2$ adsorption capacity both decrease as the current density increases, but with different decreasing rates and different profiles observed (Fig. 2g and Supplementary Fig. 5). While the capacitance shows an almost linear decrease of 15% between 10 and 150 mA g$^{-1}$ as the current density increases, the $CO_2$ adsorption capacity shows a larger decrease of 41%, suggesting that the $CO_2$ capture process is slower than the energy storage process. The $CO_2$ adsorption capacity also exhibits a multi-stage profile as the current density changes, with an obvious decrease seen from 30 to 100 mA g$^{-1}$, and apparent "plateau" regions seen at extreme currents of below 30 mA g$^{-1}$ and above 100 mA g$^{-1}$. Moreover, the electrical energy consumption increases with increasing current density due to (i) the decrease in the $CO_2$ adsorption capacity and (ii) the increase in the current needed during the voltage hold to balance the polarization effect (Supplementary Fig. 6)[23].

To explore the system behaviors at different current densities further, $CO_2$ pressure data was examined at low and high current densities (Fig. 2h). At low current densities, the pressure change follows the voltage change in a timely manner (Fig. 2h), with the pressure plateaus observed before the voltage hold. This suggests that at a low current density, there is sufficient time for complete $CO_2$ adsorption to take place. In contrast, an obvious delay in the pressure maxima and minima occurs at the faster charging rate of 150 mA g$^{-1}$. This supports the idea that at high current densities, the electrochemical $CO_2$ capture process becomes kinetically limited. We propose that there will be a competition between charge storage involving $SO_4^{2-}$ ions from the electrolyte, and $CO_2$-derived bicarbonate ions. Compared to $SO_4^{2-}$ ions, $CO_2$-derived $HCO_3^-$ ions have a larger Stokes radius ($HCO_3^-$ ion: $2.19 \times 10^{-10}$ m, $SO_4^{2-}$ ion: $1.15 \times 10^{-10}$ m (water, 298 K)) and a smaller diffusion coefficient[24], while $CO_2$ dissolution may also become rate-limiting at high current densities[25,26], both of which may account for the decreasing $CO_2$ adsorption capacities at high charging rates. Overall, our findings show the need to optimize both the voltage holds and current densities when improving an electrochemical $CO_2$ capture process with supercapacitors.

## Effects of electrode structure on electrochemical $CO_2$ capture

Having quantified the impacts of different charging protocols, we turned to the question of how electrode structure affects electrochemical $CO_2$ capture performance. First, three types of carbon electrodes with different pore structures (Supplementary Table 1) and similar functional groups (Supplementary Table 2 and Supplementary Fig. 7) were selected to test the effects of the electrode porosity. From $N_2$ sorption analysis (Fig. 3a), the activated carbon cloth samples ACC-10 and ACC-20 show Type I isotherms consistent with predominantly microporous structures with pores smaller than 2 nm in diameter (see Supplementary Fig. 8 for pore size distributions). On the other hand, the activated carbons YP50F and YP80F show a combination of Type I and Type II or Type IV sorption profiles (Fig. 3a), indicating the dominance of micropores as well as the existence of mesopores (2–50 nm), which is also evidenced by pore size distribution analysis (Supplementary Fig. 8). Finally, CMK-3 shows a predominantly Type IV isotherm consistent with a mesoporous material (Fig. 3a and Supplementary Fig. 8). Among all the studied carbons, YP80F has the highest Brunauer-Emmett-Teller (BET) surface area and total pore volume (Supplementary Table 1).

Having characterized the porosity of the five carbon materials, we carried out electrochemical $CO_2$ capture measurements (negative charging mode, 5-min voltage holds) to assess the impacts of the pore structure and electrochemical capacitance on performance. The five-carbon materials show clear differences in their electrochemical capacitances (i.e., their abilities to store charge at a given voltage), with a range of 78 to 116 F g$^{-1}$ at the low current density (i.e., 10 mA g$^{-1}$) and a range of 68 to 97 F g$^{-1}$ at the high current density (i.e., 150 mA g$^{-1}$) (Fig. 3b). Strikingly, the $CO_2$ adsorption capacities broadly follow the same pattern as the capacitances but show much larger variations (Fig. 3c), with a range of 16 to 82 mmol$_{CO2}$ kg$^{-1}$ at the low current density (i.e., 10 mA g$^{-1}$) and a range of 8 to 44 mmol$_{CO2}$ kg$^{-1}$ at the high current density (i.e., 150 mA g$^{-1}$). Moreover, while YP50F and ACC-20 have very similar capacitances, they show clear differences in their $CO_2$ adsorption capacities (Fig. 3b and c). The mesoporous carbon CMK-3 is an interesting example that has very low $CO_2$ adsorption capacities at all current densities, despite its reasonable capacitances. Together these findings suggest that the electrochemical capacitance is an important factor in determining the electrochemical $CO_2$ capture capacity, but that other factors such as the pore size distribution also play an important role.

The rate dependencies of the $CO_2$ adsorption capacities reveal further differences among the various carbons (Fig. 3c). While the purely microporous carbon cloths (i.e., ACC-10 and ACC-20) show a monotonous decrease in $CO_2$ adsorption capacities as the charging rate is increased, the YP carbons with both micro- and mesopores show a sigmoidal trend (Fig. 3c). The plateau region of the $CO_2$ adsorption capacity at low charging rates for the YP carbons suggests that the maximum possible $CO_2$ adsorption capacities have been reached for these materials, while the lack of a plateau at low current densities for ACC-10 and ACC-20 suggests that electrochemical $CO_2$ capture remains kinetically limited. Moreover, the comparison of the pressure curves of YP80F and ACC-20 at the high current density (i.e., 150 mA g$^{-1}$) reveals obvious differences with the $CO_2$ pressure reaching a peak for YP80F within the 5-min voltage hold region, while the $CO_2$ pressure curve for ACC-20 shows a delay (Supplementary Fig. 9). A likely explanation for these phenomena is that the presence of mesopores for YP50F and YP80F enables the more rapid transport of $CO_2$-derived species in the electrode porosity[27,28]. Overall, the best-performing carbon is YP80F with a $CO_2$ adsorption capacity of 81 mmol$_{CO2}$ kg$^{-1}$ at 30 mA g$^{-1}$, which is increased to 170 mmol$_{CO2}$ kg$^{-1}$ with the switching charging protocol (Supplementary Fig. 10). On the other hand, the predominantly mesoporous material CMK-3 shows low $CO_2$ adsorption capacities at all current densities. Together these findings suggest that mesopores facilitate $CO_2$ transport but that a significant

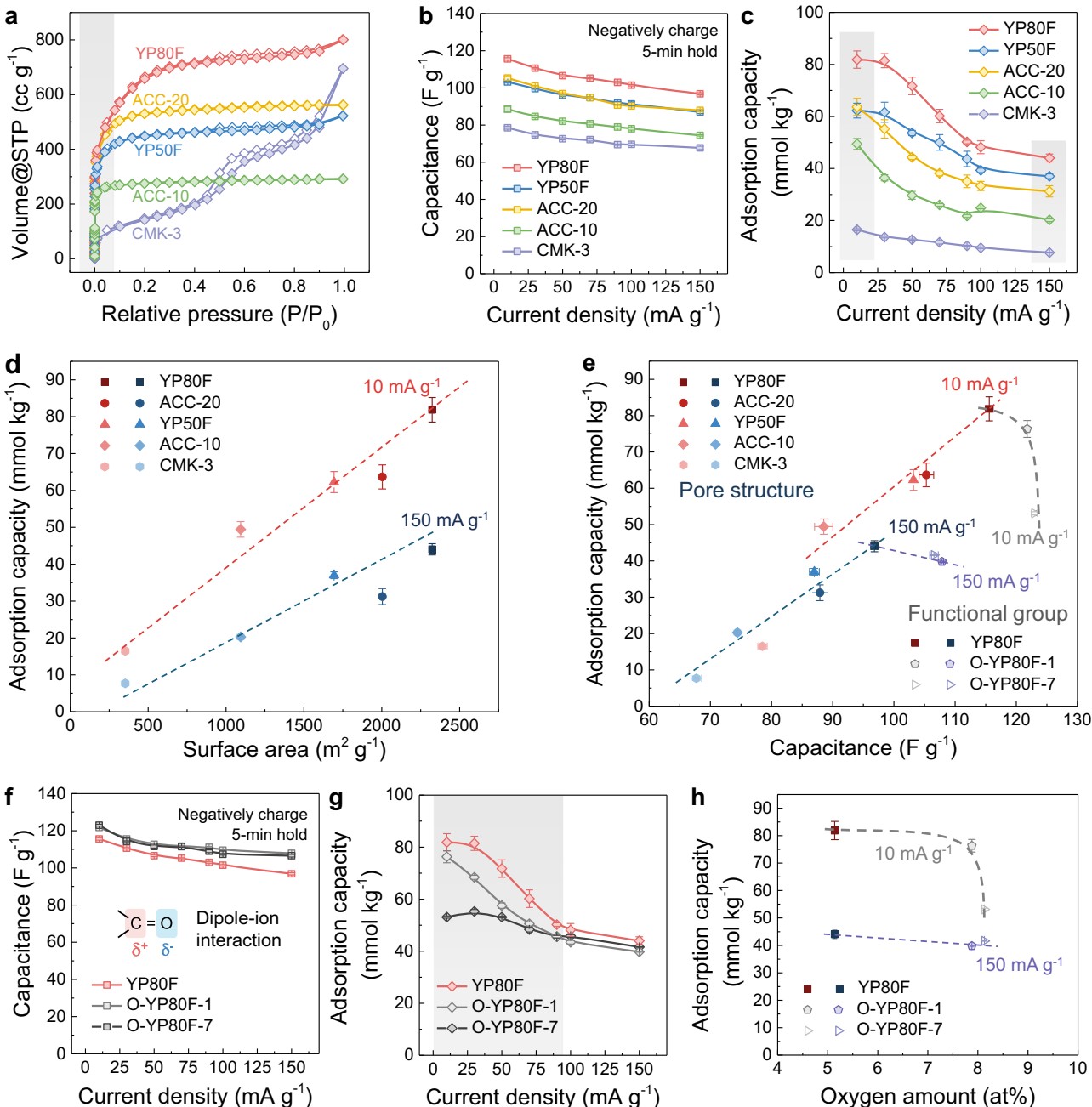

**Fig. 3 | Effects of electrode structure on electrochemical CO₂ capture. a** N₂ sorption isotherms at 77 K of carbon electrode materials (YP50F, YP80F and CMK-3 in the pristine powder form, ACC-10 and ACC-20 in the pristine cloth form) with different pore structures (filled and hollow symbols show adsorption and desorption respectively, and the gray region represents the isotherm related to micropore structures). Comparison of **b** the discharge capacitances and **c** CO₂ adsorption capacities of carbon electrodes with different pore structures under CO₂ at different current densities in the negative charging mode, with 5-min voltage holds (gray regions highlight the performance metrics at 10 and 150 mA g⁻¹). **d** The correlation between the surface areas and CO₂ adsorption capacities (at 10 and 150 mA g⁻¹) of carbon electrodes. **e** The correlation between the discharge capacitances (at 10 and 150 mA g⁻¹) and CO₂ adsorption capacities (at 10 and 150 mA g⁻¹) of carbon electrodes. Comparison of **f** the discharge capacitances and **g** CO₂ adsorption capacities of carbon electrodes with different degrees of oxidation under CO₂ at different current densities in the negative charging mode, with 5-min voltage holds (the gray region highlights the performance metrics at low current densities). **h** The correlation between the functional group amount and CO₂ adsorption capacities (at 10 and 150 mA g⁻¹) of carbon electrodes. Error bars represent the *t*-test of performance from cycle to cycle at the same charging protocol.

micropore population is still required to obtain large electrochemical CO₂ capture capacities.

Plots of the CO₂ adsorption capacity against the BET surface area show an apparent linear correlation at the low current density (*i.e.*, 10 mA g⁻¹) (Fig. 3d), while the correlation is more distorted at the high current density (*i.e.*, 150 mA g⁻¹) due to the purely microporous carbons (*i.e.*, ACC-20). When correlated with the capacitances, the CO₂

adsorption capacities show similar trends (Fig. 3e). Both plots indicate the importance of high BET surface areas and the presence of mesopores for achieving large CO₂ adsorption capacities under thermodynamic (slow charging) and kinetic (fast charging) conditions, respectively.

To explore possible differences in the transport of CO₂-derived species in the different electrodes, we performed ¹³C nuclear magnetic

resonance (NMR) spectroscopy experiments on ACC-20 and YP80F electrodes soaked with $NaH^{13}CO_3$ (aq) electrolyte as a model system to probe bicarbonate dynamics. The spectra for both samples show in-pore and ex-pore bicarbonate peaks, similar to previous NMR studies of ion adsorption in porous carbons (Supplementary Fig. 11)[29–32], and 2D exchange spectroscopy measurements confirmed that these environments undergo chemical exchange (Supplementary Fig. 12)[33]. Interestingly, the $^{13}C$ NMR spectrum of YP80F shows a much broader ex-pore peak than ACC-20, suggestive of faster $H^{13}CO_3^-$ ion exchange between the in-pore and ex-pore environment[34]. These findings provide support for the importance of $H^{13}CO_3^-$ ion exchange in explaining the faster $CO_2$ adsorption observed in YP carbons. This faster ion exchange likely stems from the presence of mesopores in YP80F, which are absent in ACC-20, though we cannot completely rule out additional effects arising from the different morphologies of these two kinds of samples (Fig. 2b and Supplementary Fig. 13).

Having explored the impacts of the electrode porosity, we next explored the impacts of oxygen functional groups, as these are well known to influence the electrochemical characteristics of aqueous supercapacitors[35]. YP80F, our best-performing carbon, was oxidized following a literature procedure using an aqueous hydrogen peroxide ($H_2O_2$) solution (see "Methods")[36]. Gas sorption measurements confirm that there are minimal changes in the BET surface area and pore size distribution upon oxidation (Supplementary Table 1 and Supplementary Fig. 14), while X-ray photoelectron spectroscopy (XPS) analysis confirms an increase in the oxygen atomic amount on the surface of O-YP80F-1 (a sample oxidized for 1 day) and O-YP80F-7 (a sample oxidized for 7 days), by approximately 2.5% and 3.0%, respectively (Supplementary Table 2 and Supplementary Fig. 15).

The oxidized YP80F demonstrates increased hydrophilicity compared to the pristine YP80F, as indicated by contact angle measurements (Supplementary Fig. 16). In addition, the cyclic voltammetry (CV) curve of oxidized YP80F exhibits broad redox peaks (Supplementary Fig. 17), suggesting improved capacitance derived from pseudocapacitive interactions between electrolyte ions and oxygen functional groups[37]. Moreover, electrochemical impedance spectroscopy (EIS) measurements support lower charge-transfer resistance at the interface of oxidized YP80F than pristine YP80F (Supplementary Fig. 17), which can be attributed to the enhanced wettability and ion-dipole interactions caused by the additional oxygen functional groups[37]. Oxidized YP80F also shows a very similar peak shape and width to YP80F in NMR spectra (Supplementary Fig. 11), indicating similar $H^{13}CO_3^-$ ion exchange rates and pore environments for both carbons.

The two oxidized YP80F samples exhibit higher capacitances than YP80F at $10\ mA\ g^{-1}$, and this difference persists as the current density increases (Fig. 3f). In contrast, the $CO_2$ adsorption capacities of oxidized YP80F samples are consistently lower than that of YP80F at low current densities (Fig. 3g), indicative of the negative effect of oxygen functional groups on the thermodynamic performance of electrochemical $CO_2$ capture (Fig. 3h). This again shows that while in general, we see the correlation between the electrochemical capacitances and $CO_2$ adsorption capacities, other factors also play an important role (Fig. 3e). A full mechanistic study of the origin of the detrimental effects of oxygen functional groups is beyond the scope of this study, but we propose that changes to the relative binding strengths of $SO_4^{2-}$ anions and $CO_2$-derived anions with the carbon surface may play an important role[38,39]. For a detailed study on the effect of oxygen functional groups on the carbon surface on the affinity of ions, please refer to our recent study[40].

Summarizing, our findings suggest that high-performing carbon electrodes for electrochemical $CO_2$ capture can be developed by designing activated carbons with a high surface area, a combination of micro- and meso-pores, and a low amount of oxygen functional groups.

## The potential for practical $CO_2$ capture applications

After establishing YP80F as our best-performing electrode material, we examined the potential of this material for practical $CO_2$ capture applications. A key parameter for real applications is the $CO_2$ adsorption rate (i.e., the adsorption capacity per unit time). With a 5-min voltage hold step, the $CO_2$ adsorption rate of YP80F plateaus with increasing current densities, stabilizing at around $300\ mmol_{CO_2}\ kg^{-1}\ h^{-1}$ (Fig. 4a and Supplementary Fig. 18). Excitingly, we find that the removal of the voltage hold step leads to a larger adsorption rate exceeding $350\ mmol_{CO_2}\ kg^{-1}\ h^{-1}$ (Fig. 4b). The decreased $CO_2$ adsorption capacities in the absence of a voltage hold (Fig. 4b and Supplementary Fig. 19) are more than counterbalanced by the decrease in the charge-discharge time for each cycle. Our measured adsorption rate is comparable with previously measured values for similar devices, together with a comparable volumetric $CO_2$ adsorption capacity (Supplementary Table 3). In addition, we note that an unexpected decrease in $CO_2$ adsorption capacity was observed at ultra-low current densities (Fig. 4a and b), suggesting a competition between adsorption and desorption effects at these conditions. We propose that $CO_2$ adsorption at the gas-exposed working electrode is accompanied by desorption at the electrolyte-immersed counter electrode, with these effects only becoming apparent at slow charging conditions. A full mechanistic study of this effect is ongoing in our laboratory.

In pursuit of developing economically viable $CO_2$ capture technologies, energy efficiency is paramount. Remarkably, the removal of the voltage hold steps greatly improves the energy efficiency as most of the irreversible electrical energy consumption originates from the voltage hold period at the charged state (Fig. 4a and b). We observe very low electrical energy consumption below $3\ kJ\ mol_{CO_2}^{-1}$ at $30\ mA\ g^{-1}$, though note that this is for capture and release under pure $CO_2$ conditions. Moreover, the average electrical energy consumption remains below $20\ kJ\ mol_{CO_2}^{-1}$ even at fast charging conditions ($300\ mA\ g^{-1}$) where the $CO_2$ adsorption rate is maximized (Fig. 4b). The low electrical energy consumption values arise from the very small cell voltage differences between charging and discharging (Supplementary Fig. 19), underscoring a key advantage of supercapacitors for electrochemical $CO_2$ capture compared to more battery-like[8] or catalytic approaches[15] (Supplementary Table 3). In short, the promising adsorption rates and low electrical energy consumption values show the potential of YP80F electrode-based supercapacitors for electrochemical $CO_2$ capture applications.

After assessing the system performance under pure $CO_2$ conditions, we turned to the question of $CO_2$ selectivity over other flue gas components. First of all, cyclic voltammetry was performed under pure $CO_2$, $N_2$, and $O_2$ (Fig. 4c). Under $N_2$, a purely capacitive CV curve is observed as expected, regardless of the voltage polarity (Fig. 4c). Consistent with the CV curve, the YP80F-based supercapacitor device shows no electrochemical $N_2$ sorption in both negative and positive charging modes (Fig. 4d and Supplementary Fig. 20). This mirrors previous work that also showed good selectivity for $CO_2$ over $N_2$ by similar types of supercapacitors[16]. The absence of $N_2$ adsorption can be attributed to the inability of $N_2$ to be converted into ions in the aqueous electrolyte, and the lack of affinity for molecular $N_2$ to carbon surfaces at room temperature. These findings are consistent with the idea that $CO_2$ capture and release by these systems are driven by perturbations in the carbonate equilibria during charging and discharging.

We next examined the system performance in the presence of $O_2$, a first for this technology. While $O_2$ is typically present at a level of 3 to 5 vol% in industrial flue gases[41], we first tested the system in pure $O_2$ at ~1 bar to explore the limits of the system. The cyclic voltammogram under $O_2$ shows clear faradaic peaks in the negative charging mode (i.e., when the gas-exposed working electrode carries electrons) (Fig. 4c), which is related to irreversible oxygen reduction reactions and possible electrode oxidation processes (Supplementary Fig. 1)[42].

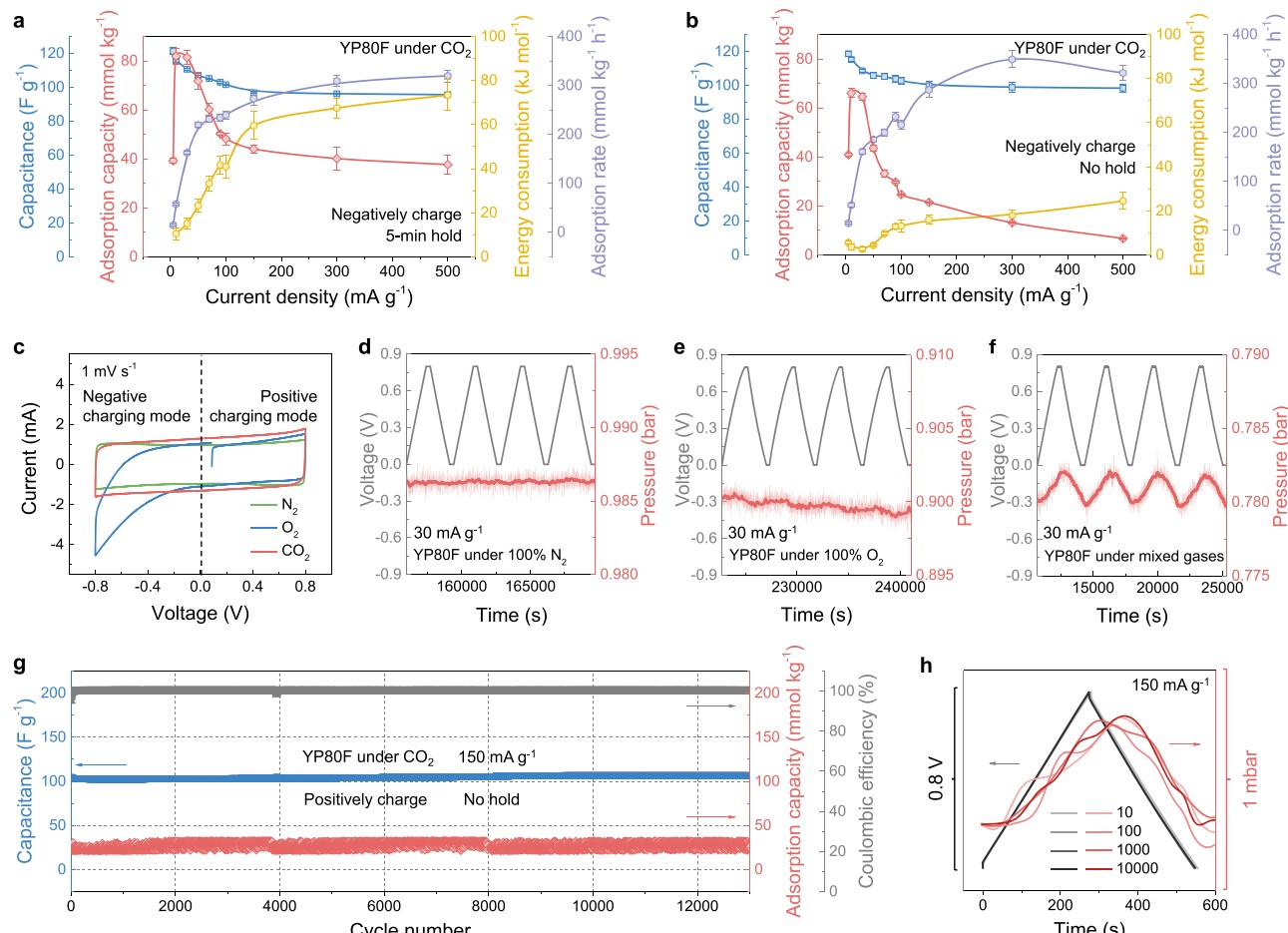

**Fig. 4 | The demonstration of the potential for practical $CO_2$ capture applications. a** Comparison of the discharge capacitance, $CO_2$ adsorption capacity, electrical energy consumption and adsorption rate (normalized by charging time) of the YP80F electrode under $CO_2$ at different current densities of 5, 10, 30, 50, 70, 90, 100, 150, 300 and 500 mA g$^{-1}$ in the negative charging mode, with 5-min voltage holds. **b** Comparison of the discharge capacitance, $CO_2$ adsorption capacity, electrical energy consumption and adsorption rate (normalized by charging time) of the YP80F electrode under $CO_2$ at different current densities of 5, 10, 30, 50, 70, 90, 100, 150, 300 and 500 mA g$^{-1}$ in the negative charging mode, without voltage hold. **c** CV curves of the YP80F electrode from −0.8 to +0.8 V at the scan rate of 1 mV s$^{-1}$ under pure $N_2$, $O_2$ and $CO_2$. Zoomed GCD curves (grey), original pressure curves (light red) and smoothed pressure curves (averaged every 100 sec, dark red) of the YP80F electrode under **d** $N_2$, **e** $O_2$ and **f** a mixed gas atmosphere of approximately 20% $CO_2$, 15% $O_2$, and 65% $N_2$ (~0.8 bar total pressure) at the current density of 30 mA g$^{-1}$ in the positive charging mode, with 5-min voltage holds. **g** Long cycling performance under $CO_2$, including the discharge capacitance, $CO_2$ adsorption capacity and Coulombic efficiency of the YP80F electrode at the current density of 150 mA g$^{-1}$ in the positive charging mode, without voltage hold (Notes: Two distortions of the $CO_2$ adsorption capacity curve were caused by the unexpected power outage in our Chemistry Department, which also explored the system stability under an external interference). **h** The 10th, 100th, 1000th and 10000th cycles of zoomed GCD curves and smoothed pressure curves (averaged every 100 sec) of the YP80F electrode under $CO_2$ during cycling. Error bars represent the $t$-test of performance from cycle to cycle at the same charging protocol.

Promisingly, these faradaic processes are suppressed in the positive charging mode (*i.e.*, when the electrolyte-immersed counter electrode carries electrons), suggesting that oxygen reduction reactions at the counter electrode may be kinetically limited under these conditions[42]. Consistent with these observations, the YP80F electrode-based supercapacitor device shows minimal irreversible pressure decrease in the positive charging mode under pure $O_2$ (Fig. 4e, Supplementary Figs. 21 and 22). The average Coulombic efficiencies under pure $O_2$ at 30 mA g$^{-1}$ and 100 mA g$^{-1}$ exceed 85% and 94%, respectively, which are already comparable to values reported for other electrochemical $CO_2$ capture methods under more dilute $O_2$ conditions. For example, Coulombic efficiency values of 65–82% were obtained under 20% $O_2$ gas flow for direct $CO_2$ capture from the air using pH-swing approaches[6,43], and values of 87–95% were obtained under 3% $O_2$ gas flow for post-combustion $CO_2$ capture from flue gases using pH-swing approaches or redox-active $CO_2$ binding molecules[6,8].

Indeed, large irreversible pressure decreases are accompanied by reversible $O_2$ pressure changes in the negative charging mode (Supplementary Figs. 21 and 22). This suggests that irreversible $O_2$ consumption occurs in these conditions, alongside partially reversible $O_2$ capture at the electrode surface[44,45]. According to the CV curve (Fig. 4c), the irreversible $O_2$ consumption is correlated with the irreversible oxygen reduction reactions. EIS experiments also support the existence of faradic reactions under $O_2$ at the negatively charged state (Supplementary Fig. 23). However, the detailed mechanism and generality of the reversible uptake of $O_2$ require further investigation as there is no obvious oxygen evolution reaction peak in the CV curve (Fig. 4c)[42]. Crucially, the flexibility of our supercapacitor devices to operate in either positive or negative charging modes provides a clear strategy to minimize oxygen side reactions–namely operating in the positive charging mode.

To explore the $CO_2$ adsorption selectivity under more realistic conditions, electrochemical $CO_2$ capture measurements were conducted under mixed gas conditions in the positive charging mode. With a gas mixture of approximately 20% $CO_2$, 15% $O_2$ and 65% $N_2$, YP80F exhibits clear reversible electrochemical adsorption behavior

(Fig. 4f). Assuming the reversible pressure changes arise from $CO_2$ alone (as suggested by Fig. 4d and e), we obtained a $CO_2$ adsorption capacity of 83 $mmol_{CO_2}$ $kg^{-1}$ and a discharge capacitance of 115 F $g^{-1}$ with a Coulombic efficiency over 99.8% at 30 mA $g^{-1}$ (Fig. 4f and Supplementary Fig. 24). Excitingly, the electrochemical $CO_2$ capture performance of YP80F under this gas mixture is very similar to that under pure $CO_2$ (Supplementary Fig. 10), and the high Coulombic efficiency value of over 99.8% is superior to those of previously reported approaches ranging from 65 to 95% under similar gas mixtures[6,8,43]. With varied current densities and the removal of the voltage hold, we observe a minor kinetic effect of the gas mixture on $CO_2$ capture performance, where a maximum adsorption rate of 310 $mmol_{CO_2}$ $kg^{-1}$ $h^{-1}$ is observed at 100 mA $g^{-1}$ (Supplementary Figs. 25, 26 and 27), which is slightly poorer than the maximum adsorption rate of 350 $mmol_{CO_2}$ $kg^{-1}$ $h^{-1}$ under pure $CO_2$ conditions. Nevertheless, the low electrical energy consumption values ranging from 0.3 to 12 kJ $mol_{CO_2}^{-1}$ under mixed gases are exceptional, outperforming other reported electrochemical $CO_2$ capture technologies (Supplementary Table 3). To further explore the minimum $CO_2$ concentration required for effective electrochemical $CO_2$ capture, we also conducted the measurement under ambient air conditions at 0.8 bar (~400 ppm $CO_2$) and found no significant reversible $CO_2$ uptake within our detection limits (Supplementary Fig. 28). This suggests that the current system and operational setup are more suitable for post-combustion $CO_2$ capture scenarios, where $CO_2$ concentrations are higher, such as those found in industrial flue gases. However please note that a limitation of our study is that we capture and release $CO_2$ from and into the same gas mixture, rather than performing a gas separation. Future experiments will use static gas or flow gas measurements in a batch mode to capture and concentrate $CO_2$ from mixtures, as in other studies (Supplementary Fig. 29)[18,46]. The promising $CO_2$ selectivity over $N_2$ and $O_2$ is a significant advantage of the supercapacitor system as $O_2$ is typically considered as a "toxic" component for many $CO_2$ capture technologies that severely compromises the separation selectivity[15].

Furthermore, few studies have addressed the operational lifetime of electrochemical $CO_2$ capture devices, and stability remains a major challenge[6]. To test the long-term stability of our system, we first carried out prolonged cycling tests at 150 mA $g^{-1}$ in pure $CO_2$ conditions. After 12000 cycles (over 2500 h of operation), no noticeable fading is observed in discharge capacitance, Coulombic efficiency or $CO_2$ adsorption capacity (Fig. 4g and h). Even at a smaller current density (100 mA $g^{-1}$) with a 5-min voltage hold, no noticeable fading is observed after 1000 cycles (Supplementary Fig. 30). Importantly, this is also true of cycling tests under mixed gas conditions (~20% $CO_2$, 15% $O_2$ and 65% $N_2$) after 1000 cycles (Supplementary Fig. 31). Our results contrast with the cycling performance of other reported electrochemical $CO_2$ capture approaches, such as using redox-active quinone capture agents, which showed around 50% loss in $CO_2$ capture capacity over 200 cycles under pure $CO_2$[9], or around 30% loss in $CO_2$ capture capacity over 7000 cycles under pure $CO_2$ with polymerized quinones[7]. The excellent robustness of our YP80F electrode-based supercapacitor system further underscores its promise for electrochemical $CO_2$ capture applications.

Finally, we note that the YP80F electrode-based $CO_2$ capture system is highly sustainable as the system employs biomass-based carbon materials and low-cost aqueous electrolytes, and the electricity required to drive the process can be generated from renewable energy sources. This approach has the possibility to minimize the environmental impact and the system's carbon footprint during both fabrication and operation. A life cycle assessment is needed to quantitatively analyze the carbon footprints of the studied system in the future[47]. A preliminary techno-economic analysis on supercapacitive swing adsorption has also recently been published elsewhere[18].

In summary, the systematic exploration of the YP80F electrode's performance in terms of adsorption rate, energy efficiency, selectivity, stability and sustainability provides a comprehensive assessment of its potential for practical electrochemical $CO_2$ capture applications.

## Discussion

This work has presented a detailed study of the impacts of electrode structure and charging protocols on electrochemical $CO_2$ capture by aqueous supercapacitors. In terms of charging protocols, we have found that the use of short voltage holds, or the removal of these entirely, gives rise to the best energy efficiencies and $CO_2$ capture rates. By studying a series of activated carbons with a range of porosities, we find that carbons with large BET surface areas and large electrochemical capacitances have the highest electrochemical $CO_2$ capture capacities. At high charging rates, a combination of micro- and meso-pores is essential to achieve high $CO_2$ capacities. Meanwhile, the oxidation of the porous carbon leads to lower $CO_2$ capacities despite increases in electrochemical capacitances. The biowaste-derived activated carbon, YP80F, with a high BET surface area, a combination of micro- and meso-pores and low oxygen functionalization shows the best electrochemical $CO_2$ capture among the studied carbons. Importantly, while oxygen reduction reactions can occur on the negatively charged electrode, we show that this issue can be greatly mitigated by operating in a positive charging mode, a unique advantage of supercapacitors compared to other electrochemical $CO_2$ capture systems. The demonstrated adsorption rate, energy efficiency, selectivity and stability, especially in the presence of $O_2$, highlight the promise of electrochemical $CO_2$ capture with supercapacitors. The remaining challenges include the optimization of $CO_2$ separation from realistic flue gases, prevention of electrolyte evaporation, improvement of $CO_2$ capture performance, and stacked cell design and scale-up for industrial usage. Overall, our work will guide the design of improved supercapacitor electrodes and charging protocols for electrochemical $CO_2$ capture towards practical applications.

## Methods
### Carbon electrode fabrication
Electrodes were prepared using activated carbons and polytetrafluoroethylene (PTFE) binder, maintaining a 95:5 weight ratio. The hierarchical porous carbons employed were YP50F and YP80F (powder, Kuraray), as well as the mesoporous carbon CMK-3 (powder, ACS Material). For the oxidation of YP80F, 400 mg of carbon powders were mixed with 15 mL of $H_2O_2$ solution (30 % (w/w) in $H_2O$, Sigma Aldrich) under magnetic stirring for 1 day or 7 days. The resulting oxidized carbon powders were washed with deionized water for 3 times and dried in an incubator under 60 °C overnight. Before the electrode fabrication, all the carbon materials were dried in a vacuum oven at 95 °C overnight. For the electrode fabrication, the carbon materials were dispersed in 5 mL of absolute ethanol (Sigma Aldrich) and combined with a PTFE dispersion (60 wt% dispersion in $H_2O$, Sigma Aldrich), followed by stirring for roughly an hour to attain a dough-like consistency after ethanol evaporation. The mixture was subsequently rolled onto a glass sheet with a roller (0.25 mm thickness) to create a free-standing electrode. This electrode was transferred onto aluminum foil and dried in a vacuum oven at 95 °C overnight. Furthermore, a commercially available free-standing microporous carbon electrode (ACC-10 and ACC-20, cloth, Kynol) was utilized. Before that, it was washed with deionized water for 1 min and dried in a vacuum oven at 95 °C overnight. Circular electrodes with a diameter of 0.5 inches (around 12 mm) were cut out to achieve an approximate mass of 15 mg for $CO_2$ capture testing purposes.

### Material characterization
The morphologies of carbon materials were studied by scanning electron microscopy (TESCAN MIRA3 FEG-SEM) at 5 kV. The pore

structures of carbon materials were tested using $N_2$ sorption isotherms (Anton Parr Autosorb iQ-XR) at 77 K. Before the testing, samples were degassed at 120 °C under vacuum for 16 h. Brunauer–Emmett–Teller surface areas were calculated from isotherms using the BET equation, and pore size distributions were obtained using the quenched solid density functional theory (QSDFT) and slit pore model[48]. The surface chemistry of carbon materials was characterized using X-ray photoelectron spectroscopy (Thermo Fisher K-Alpha* XPS facility) with a monochromated Al-Kα X-ray source. Before the testing, samples were stuck onto the specific sample holders using conductive double-sided carbon tapes. Before the analysis of XPS, the samples were degassed under a high vacuum ($< 5 \times 10^{-7}$ bar) for 90 mins. Survey scans were measured using 200 eV pass energy, 1 eV step size and 200 ms (10 ms × 20 scans) dwell times and analyzed using the Avantage software. Atomic compositions were calculated and averaged according to the spectra acquired from 2-3 different spots on each sample. A contact angle goniometer (DSA25S, Kruss) was used to analyze the surface wettability of carbon electrodes with DI water. $^{13}C$ magic-angle spinning nuclear magnetic resonance (MAS-NMR) data were collected on a wide bore 9.4 T magnet with a Bruker NEO solid-state spectrometer, using either a 3.2 mm triple resonance probe or a 2.5 mm double resonance probe at a MAS rate of 5 kHz for each. $^{13}C$ chemical shifts were referenced to adamantane, with the left-hand resonance set at 37.77 ppm. The recycle delay time was varied according to the relaxation times $T_1$ for each sample to achieve quantitative results for 1-D spectra. NMR spectrum analysis was performed in Topspin v4.1.4.

## Electrochemical $CO_2$ capture measurements

Three-electrode measurements were performed in the Swagelok cell, as shown in Supplementary Fig. 2, where we used two identical carbon electrodes (YP80F) (Diameter: 8 mm) as the working and counter electrodes, respectively. Two GF/A separators (Whatman, Diameter: 10 mm) and 750 μL of 1 M $Na_2SO_4$ (aq) electrolyte were used. Together with the Hg/HgO reference electrode, the cyclic voltammetry was conducted to monitor the corresponding potential changes of the working and counter electrodes at the scan rate of $1\,mV\,s^{-1}$. Electrochemical gas adsorption experiments were performed using a custom-designed gas cell (Fig. 2a) at 303 K[17]. A symmetrical supercapacitor with a 1 M $Na_2SO_4$ (aq) electrolyte was assembled within a coin cell with a meshed top case to allow gas access (SS316 CR2032, Cambridge Energy Solution). During coin cell assembly, two identical carbon electrodes (Diameter: 12 mm), two 0.5 mm stainless steel spacers, one conical spring, two GF/A separators (Whatman, Diameter: 20 mm) and 200 μL of 1 M $Na_2SO_4$ (aq) electrolyte were used. After assembly, all components including electrodes in the meshed coin cell were firmly stacked together with a fixed thickness of 3.2 mm. Based on our findings, 100 μL of electrolyte is the recommended amount to fully infiltrate one piece of the separator (Supplementary Fig. 32). After that, the meshed coin cell was inserted in the gas cell with the mesh side facing the gas reservoir, followed by the filling of the gas reservoir with pure $CO_2$ (99.80% purity, BOC), $N_2$ (99.998% purity, BOC) or $O_2$ (99.5% purity, BOC). For air-to-$CO_2$ exchange in the gas reservoir, a gas manifold was employed (Supplementary Fig. 33). To prevent electrolyte evaporation, the cell was subjected to a static vacuum. Subsequently, the valve closest to the cell was shut, and the gas manifold was dosed with $CO_2$ at around 1.3 bar. The decreased pressure in the gas cell aids the mixture of the gas reservoir with $CO_2$ from the manifold upon opening the cell valve. Then the cell valve was closed, and the manifold returned to dynamic vacuum. This dosing process was iterated 4 more times to establish an approximately pure $CO_2$ headspace. The same dosing protocols were employed for air-to-$N_2$ and air-to-$O_2$ exchanges. For the introduction of mixed gases, the gas reservoir was firstly dosed with 0.5 bar pure $CO_2$, and the gas manifold was introduced with 1 bar air. By mixing approximately 17 mL of 0.5 bar pure $CO_2$ in the gas reservoir and approximately 30 mL of 1 bar air in the gas

manifold, the resulting mixed gases of approximately 20% $CO_2$, 15% $O_2$ and 65% $N_2$ were obtained with a pressure of around 0.8 bar. For the measurements under ambient air conditions, the gas cell was directly dosed with 0.8 bar ambient air conditions in the gas reservoir. A potentiostat (VSP-3e and VMP-3e, Biologic) was used to conduct the electrochemical testing of gas cells including the galvanostatic charge and discharge measurement, cyclic voltammetry and electrochemical impedance spectroscopy. The gas adsorption or desorption was measured in a 30 °C incubator (SciQuip Incu-80S) by monitoring the gas reservoir pressure of the electrochemical gas cell with a pressure transducer (PX309-030A5V, Omega). The noise of the pressure transducer is at the level of 0.1 mbar, and the signal-to-noise ratio is over 5, which indicates a reasonable sensitivity of the pressure sensor. Also, we averaged the pressure data every 100 seconds to further decrease the effect of pressure noise. The error presented in this work is dominated by the cycle-to-cycle difference rather than the noise of the pressure sensor. Additionally, we validated the pressure transducer using the two additional pressure sensors (MKS PDR2000 Dual Capacitance Manometer) on the gas manifold with accuracy at the level of 0.01 mbar (Supplementary Fig. 33), ensuring high measurement accuracy and reliability. Considering the challenges associated with equilibration time in static methods and the slower gas diffusion rates, all gas cells were pre-cycled under $1\,mV\,s^{-1}$ for 20 cycles (~8 h) during which time $CO_2$ continued to equilibrate with the cell. The 1 h rest before the regular GCD measurement was associated with a horizontal pressure baseline, which indicates the established equilibrium of the whole system after pre-cycling (Supplementary Fig. 3f). All the electrochemical $CO_2$ capture measurements were repeated using at least two independent cells to confirm the reproducibility. The detailed calculation methods can be found in the Supplementary Information.

## Data availability

The raw experimental data used in this study are available in the Cambridge Research Repository, Apollo, under accession code https://doi.org/10.17863/CAM.106678[49].

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

## Acknowledgements

This work was supported by a UKRI Future Leaders Fellowship to A. C. F. (MR/T043024/1) and an EPSRC New Horizons award (EP/V048090/1). We thank Israel Temprano, Michael De Volder, Magdalena Titirici, Christopher Truscott and Andrea Lamberti for their support of this work. We also thank Marta Santos Rodriguez and Javier Carretero-González for their feedback on this manuscript. G. M. acknowledges the NanoDTC Cambridge (EP/L015978/1) for the PhD scholarship. Z. C. acknowledges the Gates Cambridge Trust for the PhD scholarship. X. L. acknowledges

the Cambridge Trust and China Scholarship Council for the PhD scholarship. D. M. acknowledges the ERC starting grant (Grant agreement "CO2CAP" No. 949916) for the funding support.

## Author contributions

A.C.F. supervised and guided the project. A.C.F., Z.X., G.M., and Z.C. designed the research. Z.X. and G.M. completed the electrode fabrication, device assembly, electrochemical $CO_2$ capture testing, and data analysis. Z.C. operated the solid-state NMR experiments and analyzed the results. M.W. conducted the XPS measurement and analyzed the results. T.S. and Z.X. performed the SEM experiments and conducted the selectivity assessment. X.L. and Z.X. performed the $N_2$ sorption measurement and analyzed the results. D.M. and Z.X. performed the three-electrode measurement and analyzed the results. Z.X. and A.C.F. drafted the manuscript, and all authors contributed to the manuscript revision.

## Competing interests

The authors declare no competing interests.
