## [Peer Review File · Nature Communications]

Enhancing Electrochemical Carbon Dioxide Capture with SupercapacitorsREVIEWER COMMENTS

Reviewer #1 (Remarks to the Author):

The author examined how the structure of electrodes influences electrochemical carbon capture and release across various protocols and gas compositions. Through meticulous experimentation, the study reveals that electrodes featuring large surface areas, minimal oxygen functionalization, and a blend of micro- and meso-pores exhibit superior CO₂ capture capacities and rates. Furthermore, the research showcases the system's ability to attain high efficiency, stability, and selectivity for CO₂ over other gases, particularly in simulated flue gas conditions. However, the investigation into charging protocols had already been covered in a previous publication by the authors (Binford, T. B., Mapstone, G., Temprano, I. & Forse, A. C. Enhancing the capacity of supercapacitive swing adsorption CO₂ capture by tuning charging protocols. *Nanoscale* 14, 7980-7984, (2022)). Moreover, the paper fails to significantly contribute much in the field of electrochemical carbon capture. Its sole addition lies in testing the proposed system under different gases, a factor insufficient to warrant a standalone high-impact paper. To improve the research, I suggest eliminating redundant investigations such as those on charging protocols and oxygen functionality, while incorporating a study on the limitations mentioned in the manuscript such as, including gas flow measurements in batch mode to capture and concentrate CO₂ from mixtures. Given these, it is recommended that the paper go through a major revision.

Reviewer #2 (Remarks to the Author):

The work by Forse investigates a number of commercial porous carbon materials for CO₂ electrosorption. This technology is rather emerging and recently studied by Landskron. It is promising for CO₂ capture but mechanisms and parameters are not well understood. In this sense more fundamental work is required to achieve a better understanding. But also real world testing conditions need to be explored and standardized. Materials are evaluated here with respect to surface area and polarity. Also various polarity aspects and current density measurements are performed. The testing is performed with pure CO₂ and diluted. Moreover the impact of O₂ is studied. The cycling stability is quite remarkable. The fundamental insight are important.

This work is rather important and relevant. However also some testing aspects may require refinement.

Specific comments:

- How accurate is the pressure measurement? Is the delta (conc. change) only 1 mbar?
- How does the static method described here compare with the dynamic flow systems described by Landskron. It seems, equilibration is always a problem and takes time here. In the static method the gas diffusion may be quite slow. Can the authors estimate time constants?
- Mmol/kg. Is kg the mass of adsorbent? What is the impact of the electrolyte amount used in the system? Considering solubility this may be important. The volumetric efficiency is also important.
- Is it possible to estimate which CO₂ concentrations could be reached with this method in

the desorption stream? What is the minimum conc. in the incoming adsorption stream effluent for effective electrosorption? If the delta (pressure change) is only 1 mbar there may be a lower limit.

- Is it possible to confirm the proposed mechanism by experimental methods?
- What are the potentials of the electrodes vs reference electrode? Would it be useful to add 3 electrode measurements?

Reviewer #3 (Remarks to the Author):

Response to Reviewers' Comments:

Reviewer #1 (Remarks to the Author):

The author examined how the structure of electrodes influences electrochemical carbon capture and release across various protocols and gas compositions. Through meticulous experimentation, the study reveals that electrodes featuring large surface areas, minimal oxygen functionalization, and a blend of micro- and meso-pores exhibit superior CO₂ capture capacities and rates. Furthermore, the research showcases the system's ability to attain high efficiency, stability, and selectivity for CO₂ over other gases, particularly in simulated flue gas conditions.

Response: We sincerely appreciate the reviewer's recognition of our systematic and comprehensive study on the use of supercapacitors for electrochemical CO₂ capture applications.

*However, the investigation into charging protocols had already been covered in a previous publication by the authors (Binford, T. B., Mapstone, G., Temprano, I. & Forse, A. C. Enhancing the capacity of supercapacitive swing adsorption CO₂ capture by tuning charging protocols. 625 *Nanoscale* 14, 7980-7984, (2022)).*

Response: Our earlier *Nanoscale* paper mainly focused on the impact of varying voltage polarity on electrochemical CO₂ capture. However, our current study delves deeper into the effects of voltage hold time and current density, which were not investigated in our previous paper. Notably, to the best of our knowledge, this is the first study on the effect of current density for electrochemical CO₂ capture with supercapacitors since the fast charge-discharge ability is a key metric of supercapacitors. The investigation into these additional parameters is crucial for optimizing the electrochemical CO₂ capture performance of supercapacitors. By coupling the current density studies with a range of different carbons we discover the important role of combined micro and mesoporosity for high-rate electrochemical CO₂ capture.

We appreciate the reviewer's insights and hope this detailed explanation clarifies the advancements and distinctions of our current study from our previous work.

Additionally, although a similar electrochemical gas cell was reported in our previous publication (Binford, T. B., *et al. Nanoscale*, 2022, 14, 7980-7984), we have significantly advanced the device architecture in our current study to address critical issues related to reproducibility and internal resistance (**Fig. 2a**), thereby enabling the rapid screening of different materials for supercapacitive swing adsorption.

Fig. 2a. Schematic illustration of the meshed coin cell embedded in the Swagelok-type electrochemical gas cell.

In this study, we employed commercial coin cells with a meshed top case, which ensured consistent contact between the gas-exposed working electrode and the gas reservoir. This design innovation effectively mitigates the reproducibility and conductivity challenges associated with the Swagelok cell setup used in our previous *Nanoscale* paper. The internal resistance was decreased to below 2 Ω which greatly reduced the electrical energy consumption to below 20 kJ molCO_2^{-1} during the electrochemical CO_2 capture process (**Supplementary Fig. 23** and **Supplementary Table 3**). By enhancing the structural integrity and reducing internal resistances, we were able to achieve more reliable and accurate experimental outcomes.

Moreover, the paper fails to significantly contribute much in the field of electrochemical carbon capture. Its sole addition lies in testing the proposed system under different gases, a factor insufficient to warrant a standalone high-impact paper. To improve the research, I suggest eliminating redundant investigations such as those on charging protocols and oxygen functionality, while incorporating a study on the limitations mentioned in the manuscript such as, including gas flow measurements in batch mode to capture and concentrate CO_2 from mixtures. Given these, it is recommended that the paper go through a major revision.

Response: We respectfully disagree with the reviewer's comments regarding the significance of our study in the field of electrochemical CO_2 capture. Perhaps most importantly, our manuscript shows how the supercapacitor charging mode can be modified to greatly decrease oxygen reactivity, a problem that plagues most electrochemical CO_2 capture technologies (please see below). Moreover, as pointed out by Reviewer 2, we further show the excellent long-term stability of supercapacitors for electrochemical CO_2 capture, something which is not expected to be possible with more battery-like systems that utilize redox reactions to either directly capture CO_2 at a redox-active nucleophile, or to indirectly capture CO_2 via a pH swing.

Moreover, our investigations into oxygen functionality and charging protocols are far from

redundant. In fact, they are central to understanding and optimizing the performance of supercapacitors for electrochemical CO₂ capture.

(1) Oxygen functionality: As highlighted in the main text, our research aimed to distinguish the effects of surface oxygen functionalization of carbon electrodes from their pore structures. Our experiments were designed to ensure a fair comparison, providing new insights that were previously lacking in the literature. Notably, our study is the first to provide experimental evidence of the negative impact of oxygen functional groups on electrochemical CO₂ capture. This finding challenges the conventional wisdom in the field, where oxygen functional groups might be expected to be beneficial to electrochemical performance (*e.g.*, due to improved electrode wetting by the electrolyte). By elucidating this counterintuitive effect, our work advances the understanding of material properties that optimize CO₂ capture.

(2) Charging protocols: First, our investigation into the effect of current density for a range of different carbons with varying porosity is both novel and important, as we mentioned above. Also, we discovered for the first time that by manipulating the voltage polarity of the device and using a positive charging mode, we could greatly decrease O₂ reduction during the electrochemical CO₂ capture process in supercapacitors. This is because the working electrodes always carry positive charges, thus realizing excellent CO₂ selectivity over O₂ and high Coulombic efficiencies over 99.8% even under a gas mixture. This innovation unlocks the potential of supercapacitors for electrochemical CO₂ capture under the presence of O₂.

Additionally, the reviewer's recommendation to incorporate gas flow measurements in batch mode has been noted. **Supplementary Fig. 29a** now presents a schematic inspired by Liu, Y., *et al.* (*iScience*, 2022, 25, 105153), illustrating engineering designs for batch-mode electrochemical CO₂ capture. One of the designs features two valves controlling the gas inlet and outlet, enabling batch-mode electrochemical CO₂ capture in a static gas environment. We have added the new **Supplementary Fig. 29** to the supplementary information and relevant discussion to the main text, highlighted in **Green**.

On page 15 of the main text:

Future experiments will use static gas or flow gas measurements in a batch mode to capture and concentrate CO₂ from mixtures, as in other studies (Supplementary Fig. 29).^{18,46}

On page 38 of the supplementary information:

Supplementary Fig. 29. Possible engineering design of batch-mode electrochemical CO₂ capture with supercapacitors for practical applications.⁸ Schematic illustration of the loop of batch-mode electrochemical CO₂ capture with supercapacitors under (a) static gas atmosphere and (b) flow gas condition. Notes: The engineering design as shown in Supplementary Fig. 29a features two valves controlling the gas inlet and outlet, enabling batch-mode electrochemical CO₂ capture in a static gas environment.

Furthermore, it is important to recognize that several other research groups have already reported such measurements (*e.g.*, Bilal, M., *et al. Small* **2023**, 19, 2207834). In contrast, static gas atmosphere measurements, like those in our study, are less explored but provide equally valuable insights. While gas flow measurements simulate practical conditions, they can obscure essential phenomena such as corrosion and side reactions. In our static gas atmosphere setup, we observed both reversible and irreversible pressure changes, which led to the novel discovery of stainless-steel corrosion under CO₂ and carbon electrode oxidation under O₂.

We believe these advancements and the novel findings presented in our study justify its significance and impact. Nonetheless, we greatly value the reviewer's suggestions and will consider them for future work to further refine and expand our research. Thank you very much!

Reviewer #2 (Remarks to the Author):

The work by Forse investigates a number of commercial porous carbon materials for CO₂ electrosorption. This technology is rather emerging and recently studied by Landskron. It is promising for CO₂ capture but mechanisms and parameters are not well understood. In this sense more fundamental work is required to achieve a better understanding. But also real world testing conditions need to be explored and standardized. Materials are evaluated here with respect to surface area and polarity. Also various polarity aspects and current density measurements are performed. The testing is performed with pure CO₂ and diluted. Moreover the impact of O₂ is studied. The cycling stability is quite remarkable. The fundamental insight are important. This work is rather important and relevant.

Response: We thank the reviewer for their comments and recognition of our work's significance.

However also some testing aspects may require refinement.

Specific comments:

- *How accurate is the pressure measurement? Is the delta (conc. change) only 1 mbar?*

Response: Thank you for the question on this. Indeed, the reversible change in pressure is consistently observed at the mbar level in our experiments, while the noise of the pressure sensor is at the 0.1 mbar level, giving a signal-to-noise ratio above 5. We average the pressure data every 100 seconds to further decrease the effect of pressure noise. The data shown in *e.g.*, **Supplementary Fig. 3a-e** exhibits that we consistently see measurable and reproducible CO₂ pressure changes during charging and discharging, despite some visible noise in the pressure data for individual cycles. Ultimately, the CO₂ capacity values presented in this work are the average values across multiple cycles, while the errors reflect the standard deviation between different cycles (which are more significant than errors from noise in the pressure measurement).

Supplementary Fig. 3. Electrochemical CO₂ capture measurement of YP50F by varying voltage under CO₂. (a) Overall GCD curves and corresponding pressure curves of the device using YP50F electrodes for CO₂ sorption in negative, positive and switching charging modes (under CO₂, at the current density of 30 mA g⁻¹, with a 30-min voltage hold after the charge or discharge process). (b-e) Zoomed GCD curves and smoothed pressure curves (averaged every 100 sec) of the YP50F electrode under CO₂ at the current density of 30 mA g⁻¹ in different charging modes, all with 30-min voltage holds. (Negative 1 and 2 represent the first and second negative charging parts, respectively).

Additionally, we carried out repeat experiments on at least two independent cells for all the samples in this work, which confirmed the reproducibility of the results (e.g., **Supplementary Fig. 27**):

Supplementary Fig. 27. Tests of the reproducibility of electrochemical CO₂ measurements

under pure CO₂ and mixed gas conditions (approximately 20% CO₂, 15% O₂ and 65% N₂ at around 0.8 bar total pressure). Comparison of (a) CO₂ adsorption capacity and (b) the discharge capacitance of the YP80F electrode under CO₂ in the negative charging mode and mixed gas conditions in the positive charging mode at different current densities of 30, 50, 70, 90, 100 and 150 mA g⁻¹, all without voltage hold.

Furthermore, our pressure sensor undergoes validation to ensure high accuracy and reliability throughout the experimental process. As shown in **Supplementary Fig. 33** and **Figure R1**, there are two additional pressure sensors (MKS PDR2000 Dual Capacitance Manometer) with accuracy at the level of 0.01 mbar on the gas manifold which are used for the validation of the pressure sensor on the electrochemical gas cell.

Supplementary Fig. 33. The setup of the gas manifold for dosing gases. A photo of the gas manifold for dosing gases into the electrochemical gas cell, where Valve 1 was used to control the connection between vacuum pump and gas cell, Valves 2 & 3 were used to control the connection between the CO₂ cylinder and gas cell, and Valve 4 was used to control the connection between other gas cylinders and gas cell.

Figure R1. Photography of the digital screen of the pressure sensors on the gas manifold.

Last but not least, the experimental setup like the incubator oven was designed to minimize external influences and maintain stable conditions, further enhancing the accuracy of our pressure readings.

Therefore, the accuracy and reliability of our experimental results are ensured based on these methods, and we believe that this precise pressure measurement contributes to a deeper understanding of system behaviors. We have added the additional information to the experimental section, highlighted in **Yellow**.

On page 20 of the **main text**:

The gas adsorption or desorption was measured in a 30 °C incubator (SciQuip Incu-80S) by monitoring the gas reservoir pressure of the electrochemical gas cell with a pressure transducer (PX309-030A5V, Omega). The noise of the pressure transducer is at the level of 0.1 mbar, and the signal-to-noise ratio is over 5, which indicates a reasonable sensitivity of the pressure sensor. Also, we averaged the pressure data every 100 seconds to further decrease the effect of pressure noise. The error presented in this work is dominated by the cycle-to-cycle difference rather than the noise of the pressure sensor. Additionally, we validated the pressure transducer using the two additional pressure sensors (MKS PDR2000 Dual Capacitance Manometer) on the gas manifold with accuracy at the level of 0.01 mbar (**Supplementary Fig. 33**), ensuring high measurement accuracy and reliability.

All the electrochemical CO₂ capture measurements were repeated using at least two independent cells to confirm the reproducibility.

• *How does the static method described here compare with the dynamic flow systems described by Landskron. It seems, equilibration is always a problem and takes time here. In the static method the gas diffusion may be quite slow. Can the authors estimate time constants?*

Response: In comparison to the dynamic flow systems, static gas atmosphere measurements, like those we conducted, are less explored but equally critical. While gas flow measurements simulate practical conditions, they can obscure essential phenomena such as corrosion and side reactions. In our static gas atmosphere setup, we observed both reversible and irreversible pressure changes, which led to our novel discoveries of stainless-steel corrosion under CO₂ and

carbon electrode oxidation under O₂. This kind of setup can realize fast monitoring of the pressure change in the gas chamber with a reasonable sensitivity, as discussed previously, and enables the rapid screening of different materials, as was carried out in our study. This precedent demonstrates the value of static gas atmosphere studies in revealing critical insights that might be missed with flow gas setups. We believe this kind of setup fits well with the main content of this study, which is to understand the effects of charging protocols and materials properties for enhanced electrochemical CO₂ capture with supercapacitors.

We are currently building a gas flow set-up in our laboratory so that a full comparison between static gas chamber measurements and flow measurements can be carried out. However, this is ultimately beyond the scope of our current work.

We also acknowledge the potential challenges associated with equilibration time in static methods and the slower gas transport rates compared to a flow system. To allow the cell to come to equilibrium with the CO₂ gas chamber prior to carrying out electrochemical CO₂ capture experiments we conduct both pre-cycling electrochemistry experiments (~ 8 hours), as well as a period of cell resting (~ 1 hour). Our approach of conducting 20 cycles of precycling under 1 mV s⁻¹ for each cell is aimed at ensuring consistent and reliable electrochemical performance across experimental runs, while also providing time for CO₂ dissolution into the cell. This pre-treatment step helps to establish a stable pressure baseline and enhances the accuracy and reproducibility of our results (see **Supplementary Fig. 3f** for the horizontal pressure baseline during a rest period before electrochemical CO₂ capture experiments, which indicates the equilibration after precycling). The pre-cycling step takes around 8 hours.

Supplementary Fig. 3f. Zoomed GCD curves, original pressure curves (light red) and smoothed pressure curves (averaged every 100 sec, dark red) of the YP50F electrode under CO₂ during the rest before testing.

We have added the new statement on gas diffusion to the experimental section, highlighted in **Yellow**.

On page 20 of the **main text**:

Considering the challenges associated with equilibration time in static methods and the slower gas diffusion rates, all gas cells were pre-cycled under 1 mV s⁻¹ for 20 cycles (~ 8 hours) during which time CO₂ continued to equilibrate with the cell. The 1-hour rest before the regular GCD measurement was associated with a horizontal pressure baseline, which indicates the

established equilibrium of the whole system after pre-cycling (**Supplementary Fig. 3f**).

• mmol/kg. Is kg the mass of adsorbent? What is the impact of the electrolyte amount used in the system? Considering solubility this may be important. The volumetric efficiency is also important.

Response: Yes. The capacity reported in mmol kg^{-1} is normalized by the mass of the adsorbent, following the calculation methods established by Landskron's group (please see **Supplementary Equation (6)**). This normalization was chosen to allow a direct comparison with previously reported performance data.

To understand the impact of the electrolyte amount, we conducted the electrochemical CO_2 capture measurement with varying electrolyte amounts (100 μL or 200 μL of electrolyte per coin cell) as tested by two independent researchers. Researcher 1 used YP50F as the electrodes and 100 μL of 1 M Na_2SO_4 (aq.) as the electrolyte, while Researcher 2 used YP50F as the electrodes and 200 μL of 1 M Na_2SO_4 (aq.) as the electrolyte. We added the figure of the effect of electrolyte amount on electrochemical CO_2 capture into the supplementary information as the new **Supplementary Fig. 32**, highlighted in **Yellow**.

On page 41 of the **supplementary information**:

Supplementary Fig. 32. Effect of electrolyte amount on electrochemical CO_2 capture. Comparison of (a) the discharge capacitance and (b) CO_2 adsorption capacity of the YP50F electrode under CO_2 at the current density of 30 mA g^{-1} in different charging modes (*i.e.*, negative, positive and switching charging modes), with a voltage hold time of 5 min and different electrolyte amount (*i.e.*, 100 μL with one piece of the separator and 200 μL with two pieces of separators).

As shown in **Supplementary Fig. 32**, there is no significant difference in performance between the two cells.

Additionally, we have added the new discussion to the main text, highlighted in **Yellow**.

On page 19 of the **main text**:

During coin cell assembly, two identical carbon electrodes (Diameter: 12 mm), two 0.5 mm stainless steel spacers, one conical spring, two GF/A separators (Whatman, Diameter: 20 mm) and 200 μL of 1 M Na_2SO_4 (aq.) electrolyte were used. After assembly, all components including electrodes in the meshed coin cell were firmly stacked together with a fixed thickness

of 3.2 mm. Based on our findings, 100 μL of electrolyte is the recommended amount to fully infiltrate one piece of the separator (**Supplementary Fig. 32**).

We agree that the volumetric performance is a critical metric for space-limited CO_2 capture applications. Our current setup employs a commercial CR2032 coin cell with a fixed volume (Diameter: 20 mm, Thickness: 3.2 mm), limiting our ability to properly evaluate the volumetric performance of the whole device. The future development involving pouch cells or stacked cells with thinner polymer-based separators (*e.g.*, Celgard®), could enable more accurate investigation and improvement in the volumetric performance at the whole device level. At the current stage, we further calculated the volumetric performance of the working electrode according to the new **Supplementary Equation (7)**, which follows the metric calculation methods established by Landskron.

On page 3 of the **supplementary information**:

The volumetric performance is a critical metric for space-limited applications. Our current setup employs a commercial CR2032 coin cell with a fixed volume (Diameter: 20 mm, Thickness: 3.2 mm), limiting our ability to properly evaluate the volumetric performance of the whole device. The future development involving pouch cells or stacked cells with thinner polymer-based separators (*e.g.*, Celgard®), could enable more accurate investigation and improvement in the volumetric performance at the whole device level. At the current stage, the volumetric CO_2 adsorption capacity was calculated based on the gravimetric CO_2 adsorption capacity and the density of the working electrode here:

$$C_{\text{CO}_2\text{-vol}} = \frac{c_{\text{CO}_2}}{\rho} \quad (7)$$

where ρ is the density (kg L^{-1}) of the working electrode. For YP80F electrodes, the density is around 0.53 kg L^{-1} .

Meanwhile, we added a new column to **Supplementary Table 3** for comparison, highlighted in **Yellow**. The calculated volumetric performance is comparable with previous carbon-based supercapacitors for electrochemical CO_2 capture.

On page 45 of the **supplementary information**:

Supplementary Table 3. Performance comparison between various CO_2 capture technologies. Values are given as volumetric capacities, where the capacity is normalized by the volume of the gas-exposed working electrode.

	CO_2 adsorption capacity (mmol L^{-1})
SSA-1 ⁹ (15% CO_2 , 85% N_2)	40
SSA-2 ¹⁰ (15% CO_2 , 85% N_2)	133
This work (pure CO_2)	~90
This work (ca. 20% CO_2, 15% O_2, 65% N_2)	~44

• *Is it possible to estimate which CO₂ concentrations could be reached with this method in the desorption stream? What is the minimum conc. in the incoming adsorption stream effluent for effective electrosorption? If the delta (pressure change) is only 1 mbar there may be a lower limit.*

Response: Thank you for your insightful questions. Based on our measurements using a gas mixture, we observed that the absolute CO₂ amount in the gas chamber is approximately 0.1 mmol, with a composition of around 20% CO₂, 15% O₂, and 65% N₂. During each cycle of our electrochemical CO₂ capture process, it captures approximately 0.002 mmol of CO₂. This corresponds to a reduction of about 2% of the CO₂ concentration per cycle. However, this value is ultimately determined by the small size of our carbon electrodes and the large size of the gas reservoir in our experimental set-up. With larger electrodes, more significant changes to the CO₂ concentration of a given gas mixture would be anticipated.

Indeed, Bilal, M., *et al.* recently demonstrated that by stacking 12 pairs of supercapacitor-type electrochemical cells, they were able to reduce the CO₂ concentration from roughly 15% to 10% after a single capture process (*Small* **2023**, 19, 2207834). This indicates that with an optimized setup, significant reductions in CO₂ concentration can be achieved in the desorption stream, demonstrating a scalable strategy to enhance the CO₂ capture capability of supercapacitors. We added the new discussion to the supplementary information, highlighted in **Yellow**.

On page 33 of the **supplementary information**:

Based on our measurements using a gas mixture, we observe that the absolute CO₂ amount in the gas chamber is approximately 0.1 mmol, with a composition of around 20% CO₂, 15% O₂, and 65% N₂. During each cycle of our electrochemical CO₂ capture process, it captures approximately 0.002 mmol of CO₂. This corresponds to a reduction of about 2% of the CO₂ concentration per cycle. Larger reductions would be expected for cells with larger electrodes (while keeping the gas volume constant).

We recognize that there is likely a minimum CO₂ concentration in the incoming adsorption stream required for effective electrochemical CO₂ capture by supercapacitive swing adsorption. In our experiments, we also tested the electrochemical CO₂ capture under ambient air conditions at 0.8 bar (with CO₂ at ~400 ppm) and found no significant reversible CO₂ uptake within our detection limits (**Supplementary Fig. 28**). This suggests that the current system and operational parameters are more suitable for post-combustion CO₂ capture scenarios, where CO₂ concentrations are higher, such as those found in industrial flue gases. We added the new **Supplementary Fig. 28** to the supplementary information, highlighted in **Yellow**.

On page 37 of the **supplementary information**:

Supplementary Fig. 28. Electrochemical CO₂ capture measurement of YP80F by varying current under ambient air conditions at 0.8 bar. (a) Overall GCD curves and corresponding pressure curves of the device using YP80F electrodes for CO₂ sorption in the positive charging mode (under air, at the current densities of 10, 30, 50, 70, 90, 100 and 150 mA g⁻¹, without voltage hold after the charge or discharge process). (b-c) Zoomed GCD curves and smoothed pressure curves (averaged every 100 sec) of the YP80F electrode under mixed gases at the current densities of 30 and 50 mA g⁻¹ in the positive charging mode, all without voltage hold.

In brief, while our current method demonstrates the potential for CO₂ capture from higher concentration streams, practical implementation would benefit from optimizing several parameters, including electrode and system design. Further research and development are needed to refine these aspects and achieve efficient, scalable solutions for electrochemical CO₂ capture in various industrial applications.

We added the new discussion to the main text, highlighted in **Yellow**.

On page 15 of the **main text**:

To further explore the minimum CO₂ concentration required for effective electrochemical CO₂ capture, we also conducted the measurement under ambient air conditions at 0.8 bar (~400 ppm CO₂) and found no significant reversible CO₂ uptake within our detection limits (**Supplementary Fig. 28**). This suggests that the current system and operational setup are more suitable for post-combustion CO₂ capture scenarios, where CO₂ concentrations are higher, such as those found in industrial flue gases.

On page 19 of the **main text**:

For the measurements under ambient air conditions, the gas cell was directly dosed with

0.8 bar ambient air conditions in the gas reservoir.

• *Is it possible to confirm the proposed mechanism by experimental methods?*

Response: We are actively working on mechanistic studies of supercapacitive swing adsorption with multiple different approaches, including experimental spectroscopic techniques, and computational modeling. However, these studies constitute very significant bodies of work (each multiple-year project in its own right) that are beyond the scope of our present manuscript. The following mechanistic studies will be reported soon by us.

• *What are the potentials of the electrodes vs reference electrode? Would it be useful to add 3 electrode measurements?*

Response: Thank you very much for this suggestion. We have recorded new three-electrode measurements and have added these to the supplementary information as the new **Supplementary Fig. 2**, highlighted in **Yellow**.

On page 5 of the **supplementary information**:

Supplementary Fig. 2. Three-electrode measurement of symmetric supercapacitors. (a) Schematic illustration of the three-electrode Swagelok cell setup using two identical carbon working and counter electrodes together with the Hg/HgO reference electrode. **(b)** Cyclic voltammetry (CV) of the device from 0 to 0.8 V at the scan rate of 1 mV s^{-1} . The corresponding potential changes of the working electrode (red) and the counter electrode (blue) during the CV measurement *versus* **(c)** time and **(d)** current. Notes: Here the open-circuit potentials (OCP) of the working electrode and

the counter electrode are both 0.11 V *versus* the potential of the Hg/HgO reference electrode. To facilitate the visualization of equally allocated potentials between the working and counter electrodes, their varied potentials are plotted *versus* OCP.

As shown in **Supplementary Fig. 2**, the device voltage of 0.8 V is equally divided between the two electrodes (with respect to a reference) when we employed two identical carbons (YP80F) as the working and counter electrodes, respectively. We added the new discussion to the main text, highlighted in **Yellow**.

On page 6 of the **main text**:

According to the three-electrode measurement of the symmetric supercapacitor with YP carbons, the cell voltage of 0.8 V is equally divided between the two electrodes (with respect to a reference electrode (**Supplementary Fig. 2**)).

On page 19 of the **main text**:

Three-electrode measurements were performed in the Swagelok cell as shown in **Supplementary Fig. 2**, where we used two identical carbon electrodes (YP80F) (Diameter: 8 mm) as the working and counter electrodes, respectively. Two GF/A separators (Whatman, Diameter: 10 mm) and 750 μL of 1 M Na_2SO_4 (aq.) electrolyte were used. Together with the Hg/HgO reference electrode, the cyclic voltammetry (CV) was conducted to monitor the corresponding potential changes of the working and counter electrodes at the scan rate of 1 mV s^{-1} .

Reviewer #3 (Remarks to the Author):

Response: We appreciate your participation in the co-reviewing process. Thank you very much!

REVIEWERS' COMMENTS

Reviewer #1 (Remarks to the Author):

The authors comprehensively addressed all the comments raised in my original review. They also justified the importance of the study in advancing electrochemical carbon capture technologies, especially from the oxygen tolerance point of view.

They also clarified the difference between this work and the previously published investigation.

Reviewer #2 (Remarks to the Author):

All comments are carefully addressed. the work can be published.

Reviewer #3 (Remarks to the Author):

I co-reviewed this manuscript with one of the reviewers who provided the listed reports.

This is part of the Nature Communications initiative to facilitate training in peer review and to provide appropriate recognition for Early Career Researchers who co-review manuscripts